# The Influence of Sebum on Directional Reflectance of the Skin

Anna Banyś [1], Magdalena Hartman-Petrycka [1], Katarzyna Kras [1], Magdalena Kamińska [1], Beata Krusiec-Świdergoł [1], Paweł Popielski [2], Agata Lebiedowska [1,*] and Sławomir Wilczyński [1]

[1] Department of Basic Biomedical Science, Faculty of Pharmaceutical Sciences in Sosnowiec, Medical University of Silesia, 40-055 Katowice, Poland

[2] Institute of Biomedical Engineering, University of Silesia in Katowice, 39 Będzińska Street, 41-200 Sosnowiec, Poland

\* Correspondence: alebiedowska@sum.edu.pl; Tel.: +48-509923939

**Abstract:** The sebaceous glands are responsible for the secretion of sebum. Its function is to maintain a proper epidermal barrier and participate in metabolic processes within the epidermis. Excessive sebum secretion leads to the development of various seborrheic diseases. The aim of this study was to determine the in vivo correlation between the amount of sebum and the directional reflectance of the skin. Measurements were performed using a Sebumeter (Courage + Khazaka, Germany) and a directional hemispherical reflectometer (Solar 410, SOC, USA). It has been shown that the amount of sebum does not affect the directional reflectance of the skin at a wavelength of 335–380 nm. With an increase in the amount of sebum, the directional reflectance of the skin decreases at wavelengths of 400–540 nm and 480–600 nm. However, with an increase in the amount of sebum, the directional reflectance of the skin increases at wavelengths of 590–720 nm, 700–1100 nm, 1000–1700 nm, and 1700–2500 nm. The closest relationship between amount of sebum and directional reflectance of the skin was observed at a wavelength of 700–1100 nm. Reflecting/scattering radiation from the skin surface, depending on the sebum content, may be clinically significant not only in the context of exposure to solar radiation but also in the context of numerous therapeutic methods based on artificial sources of radiation. In this area, it is desirable for the radiation to penetrate the skin as effectively as possible. The obtained preliminary results confirm that the used method is an interesting alternative to spectroscopic methods.

**Keywords:** sebaceous glands; sebum; directional reflectance of the skin; visible light (VL)

## 1. Introduction

The sebaceous glands are responsible for the secretion of sebum, a greasy substance which is a mixture of free fatty acids, triglycerides, squalene, wax esters, and cholesterol and its esters. Sebum functions are to maintain a proper epidermal barrier, to participate in metabolic processes within the epidermis, to provide antibacterial and moisturizing effects, and to participate in vitamin D synthesis. Overactivity of the sebaceous glands affects the properties of the skin and leads to the development of many dermatoses [1,2].

Electromagnetic radiation, which induces many negative processes and leads to the development of cancer, is also a threat to the skin [3]. It is important to know about the factors and substances that can protect the skin against a wide range of radiation. The majority of research has focused on protection against ultraviolet light [4,5], and relatively few studies explore the harmful effects of radiation in the visible light (VL) and infrared (IR) ranges [6]. Currently known filters protect the skin against UV radiation, but they are not sufficient to protect the skin against a light beam with wavelengths in the VL and IR ranges [7,8]. In the skin, reflectance depends on its properties. The more hydrated the skin, the smoother the surface and the better the reflection of the light beam, thus the higher the reflectance value [9,10]. Sebum affects skin hydration, so it affects the reflectance value as well. The reflectance/diffuse reflection of the skin depends on its water content but also on

the smoothness of the surface. Moisturizing the skin, on the one hand, causes a decrease in the refractive index (more radiation penetrates the skin), but on the other hand, the increase in hydration causes greater smoothness of the skin. Therefore, it is to be expected that skin hydration increases specular reflectance but decreases diffuse reflectance.

With almost normal (nearly perpendicular) radiation on the skin, only a small amount is reflected due to the change in refractive index, which is 1.0 for air and about 1.55 for the stratum corneum. Therefore, in the case of perpendicularly incident radiation in the spectral range of 250–3000 nm, about 4–7% of the radiation is reflected/scattered (regardless of the melanin content in the skin). However, at an angle of incidence other than 90 degrees, as in practice, a much higher percentage of the radiation is reflected/scattered from the skin. Because the surface of the stratum corneum is not smooth and flat, the skin reflectance is not specular. The reflectance that occurs at the surface of the skin can be clinically significant. Significantly, sebum may be an important factor influencing the air–tissue reflectance/dispersion coefficient. Sebum can act not only as a combining factor but also as a "smoothing" factor for the stratum corneum [11].

Sebum is a substance that occurs naturally on the surface of the epidermis and is produced by the sebaceous glands. The main functions of sebum are to moisturize, lubricate, and protect the skin and hair, maintaining an effective hydrophobic barrier that prevents water loss and the invasion of microorganisms [12]. The composition and rate of sebum production varies from person to person and depends on where on the body sebum is produced. Sebum is a complex mixture of glycerides, free fatty acids, wax esters, and squalene. The amount and dynamics of sebum secretion change in response to numerous environmental factors, including the seasons [12,13]. Sebum production increases in summer. In addition, androgens in men stimulate especially the growth of sebum-producing sebaceous glands located primarily on the face, chest, and upper back. Systemic medications that increase sebum secretion include: testosterone, progesterone, and phenothiazine. Among the drugs that reduce the secretion of sebum, retinoids should be mentioned first [14].

The gold standard in the assessment of sebum content is the use of absorbent tapes whose optical properties change after contact with sebum. The light transmittance of the absorbent tape is measured after contact with sebum on the surface of the epidermis. The absorbent tape becomes transparent when in contact with sebum on the surface of the skin. To assess the amount of sebum, the applicator is inserted into the main unit where the transparency of the film is measured using a light source. The amount of light that passes through the tape is measured by a photocell. The light penetration represents the sebum content on the surface of the skin. The measuring surface in the case of a Sebumeter is about 64 mm$^2$, and the measurement accuracy is 5%.

The correlation between sebum content and reflectance value in the studied spectral range may have significant practical implications. First, the increased reflectance of skin with sebum vs. skin without sebum may be important in skin imaging and in the use of all kinds of radiation sources affecting the skin. This applies especially to dermatology and aesthetic dermatology. The increase in reflection coefficient of skin covered with sebum indicates that, for example, laser treatments in a given spectral range should be preceded by skin degreasing. In turn, a decrease in reflection coefficient could indicate that radiation will penetrate the skin more effectively, thus dosage should be optimized in relation to sebum content. At the same time, sebum, by scattering radiation, can change the color rendering index and thus affect all skin image acquisition techniques in the examined spectral range. Therefore, understanding the correlation between sebum content and radiation scattering factor is also important from a clinical point of view.

The aim of this study was to determine the effect of the amount of sebum on the directional reflectance of the skin. The reflectance was measured in seven spectral ranges: 335–380 nm, 400–540 nm, 480–600 nm, 590–720 nm, 700–1100 nm, 1000–1700 nm, and 1700–2500 nm.

## 2. Materials and Methods

The amount of sebum and the directional reflectance were measured at 48 points on the skin. Measurement points were set on the forehead, chin, chest, and forearm of 12 volunteers aged 23–25. All volunteers were women characterized by II and III skin phototypes. The volunteers did not sunbathe for at least 2 months before the study and did not use any cosmetics on the day of the study. The research was conducted in June 2022. The temperature and humidity of the rooms were controlled and amounted to: temperature 21 °C, humidity 40%. The choice of measurement points was associated with physiological amounts of sebum.

A Sebumeter® SM 815, part of the MPA® multi-probe adapter by Courage + Khazaka, (Koln, Germany) was used to test the amount of sebum. The Sebumeter consists of a cassette with plastic film and a measuring unit with spectrometer. The film was applied to each test point with a force of 10 N for 30 s, and then its transparency was determined using a photocell. The amount of sebum was presented in micrograms of sebum per square centimeter and ranged from 0 to 350. At each point, the Sebumeter measurement was repeated twice, and the reflectometer measurement three times. The obtained results were the average of all measurements and the subject of further analysis.

A 410-Solar Reflectometer (SOC 410 Directional Hemispherical Reflectometer) from Surface Optics Corporation (San Diego, CA, USA) was used to test the directional reflectance of the skin. It allows measurement of the reflectance values of sunlight in seven discrete spectral bands, from ultraviolet to near infrared, in the following ranges: 335–380 nm, 400–540 nm, 480–600 nm, 590–720 nm, 700–1100 nm, 1000–1700 nm, 1700–2500 nm. Sunrays that fall on the surface of the skin are reflected and absorbed in varying intensity. The reflectance coefficient indicates the amount of sunlight that is absorbed and reflected from the test point on the skin. In this study, total reflectance was analyzed. A reflectometer that works in the spectral range of 335–2500 nm, with an incidence angle of 60 degrees, was used because it faithfully reflects skin exposure to natural solar radiation. In addition, the vast majority of radiation emitters (lasers, IPL, diode lamps, and others) work in this spectral range because it covers the full range of the so-called absorption maxima for the main skin chromophores, i.e., water, hemoglobin, and melanin.

The statistical analysis was carried out using STATISTICA 13 software (Statsoft, Tulsa, OK, USA). The Shapiro–Wilk test was used to verify the normality of variable distribution. Friedman's ANOVA test and Dunn's post hoc test were used to assess the differences in directional reflectance of the skin at different wavelength ranges. The relation between amount of sebum and directional reflectance at a given wavelength range were assessed using Spearman's rank-order correlation. Significance was set at $p$ less than 0.05.

## 3. Results

The results distribution of the amount of skin sebum is shown in Figure 1. The values of the median (Me) and the quartile range (25–75%) of the amount of skin sebum were 35.50 and 6.00–66.00, respectively.

The directional reflectance of the skin significantly differed depending on the wavelength range ($p < 0.0001$) (Figure 2). The skin was characterized by the highest reflectance value at wavelength ranges 590–720 nm and 700–1100 nm; reflectance at these two ranges did not differ significantly. Compared to the reflectance values at other wavelength ranges, reflectance at 590–720 nm and 700–1100 nm was significantly higher ($p < 0.05$). The skin reflectance values at 400–540 nm and 480–600 nm did not differ significantly, but compared to the higher and lower wavelength ranges, the observed reflectance differences were significant ($p < 0.05$). The skin reflectance at wavelength range 1000–1700 nm differed significantly from the reflectance at other spectral ranges except 335–380 nm. The skin reflectance at 1700–2500 nm was the lowest and differed significantly from the reflectance at other spectral ranges except for 335–380 nm.

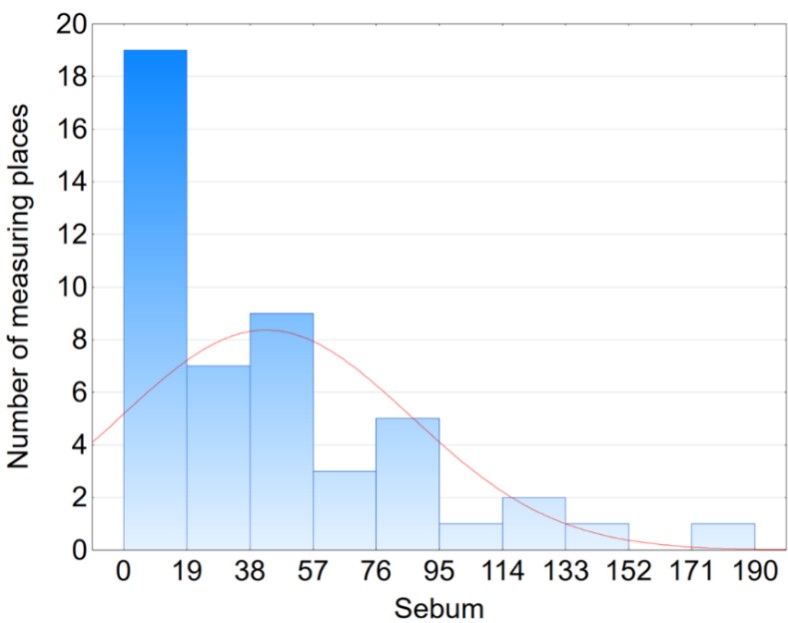

**Figure 1.** Histogram of the amount of skin sebum.

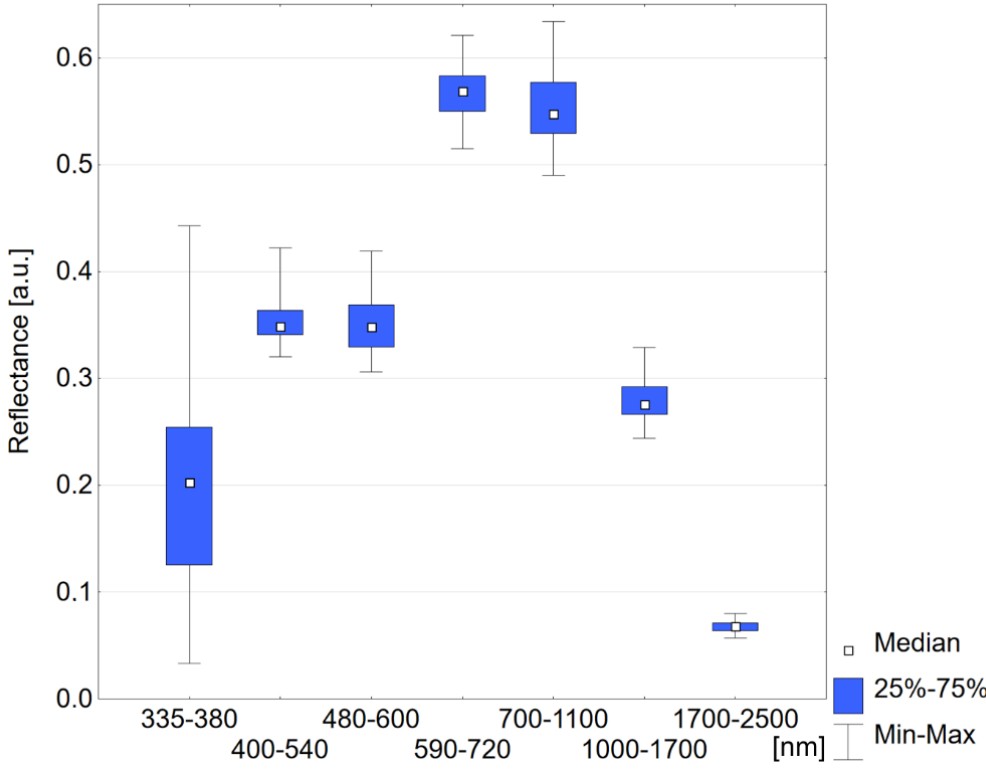

**Figure 2.** Directional reflectance values of the skin at different wavelength ranges.

Spearman's rank correlation analysis between the amount of sebum and the value of directional reflectance of the skin measured at the wavelength 335–380 nm showed no statistically significant relationship between the two variables (R = 0.14; $p$ = 0.360) (Figure 3).

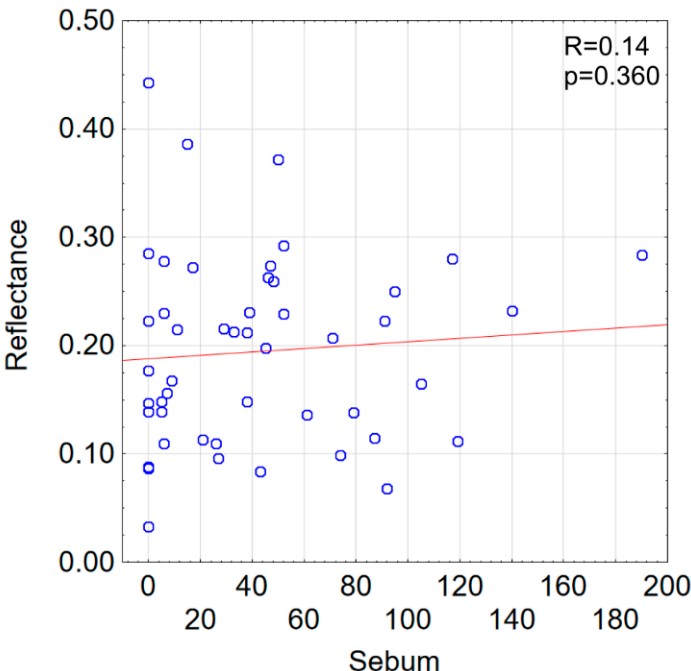

**Figure 3.** Correlation of sebum and skin reflectance at the wavelength 335–380 nm. Red line—trend line.

Negative correlations were found between the amount of sebum and the directional reflectance of the skin at the wavelengths 400–540 nm (R = −0.32; *p* = 0.026) (Figure 4) and 480–600 nm (R = −0.35 *p* = 0.014) (Figure 5).

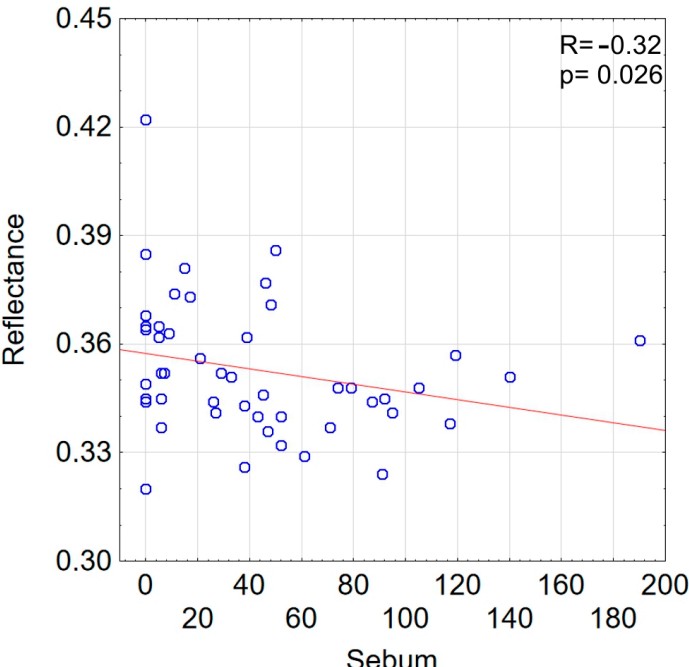

**Figure 4.** Correlation of sebum and skin reflectance at the wavelength 400–540 nm. Red line—trend line.

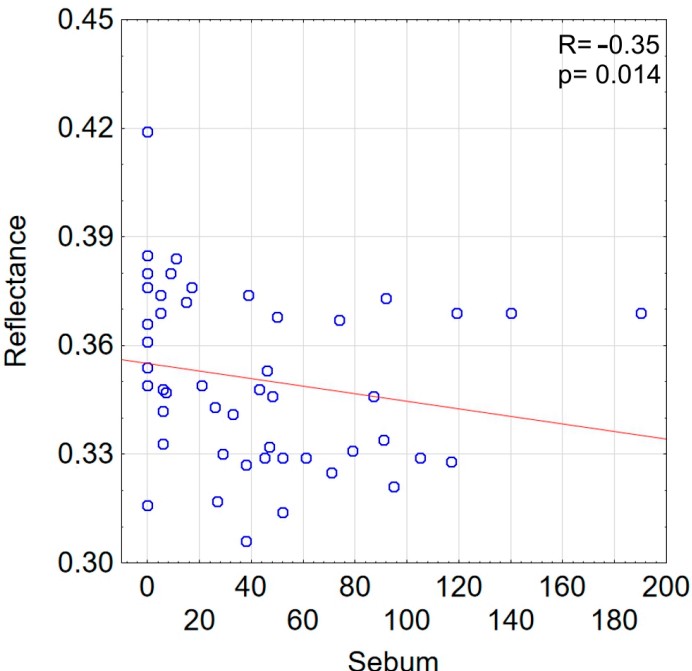

**Figure 5.** Correlation of sebum and skin reflectance at the wavelength 480–600 nm. Red line—trend line.

There was a statistically significant positive correlation between the amount of sebum and the value of directional reflectance of the skin measured at the wavelengths 590–720 nm (R = 0.35; $p$ = 0.016) (Figure 6), 700–1100 nm (R = 0.48; $p$ = 0.001) (Figure 7), 1000–1700 nm (R = 32; $p$ = 0.028) (Figure 8), 1700–2500 nm (R = 0.29; $p$ = 0.044) (Figure 9), with the strongest correlation occurring at 700–1100 nm.

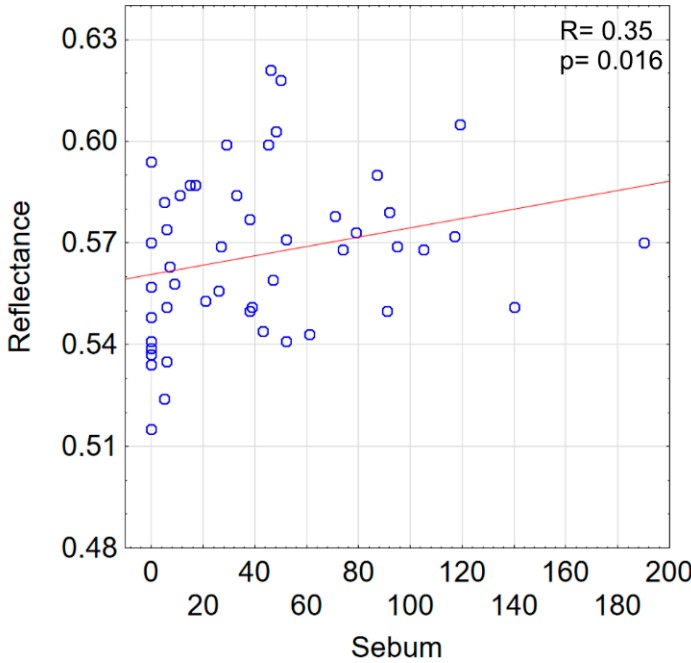

**Figure 6.** Correlation of sebum and skin reflectance at the wavelength 590–720 nm. Red line—trend line.

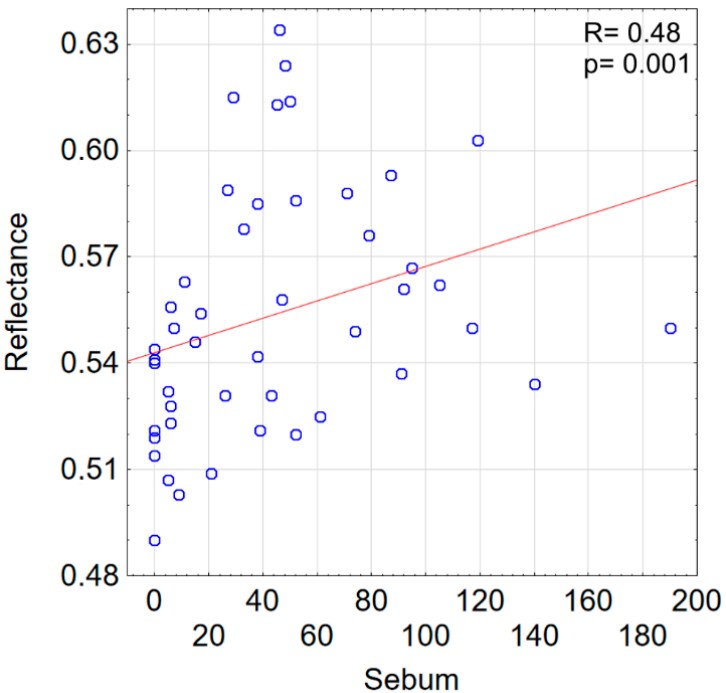

**Figure 7.** Correlation of sebum and skin reflectance at the wavelength 700–1100 nm. Red line—trend line.

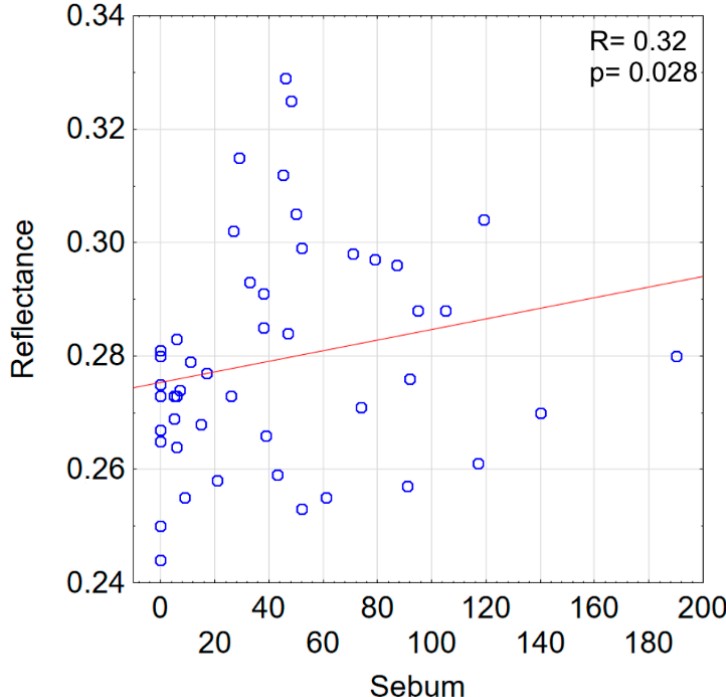

**Figure 8.** Correlation of sebum and skin reflectance at the wavelength 1000–1700 nm. Red line—trend line.

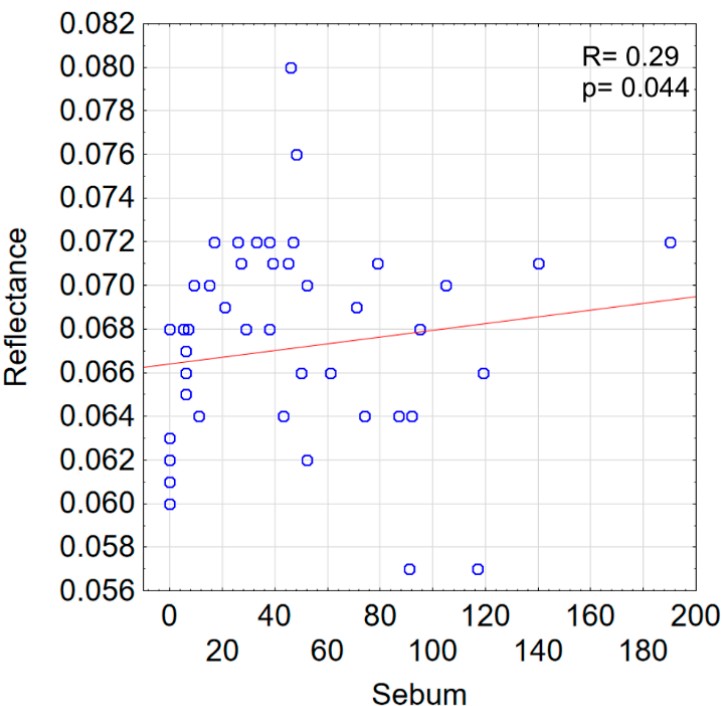

**Figure 9.** Correlation of sebum and skin reflectance at the wavelength 1700–2500 nm. Red line—trend line.

## 4. Discussion

Analysis of the correlation between the amount of sebum and directional-hemispherical reflectance was carried out. The obtained results indicate a lack of correlation in the spectral range corresponding to ultraviolet radiation (335–380 nm), while a negative correlation was obtained at the spectral ranges 400–520 nm and 480–600 nm. The lack of correlation in terms of UV radiation indicates that the amount of sebum does not have a statistically significant effect on skin reflectance. The possible reason is that in this spectral range, UV radiation is strongly absorbed by the skin (low reflectance value) and only slightly dispersed. Nevertheless, this is a surprising result if we consider that the longer the wavelength, the easier it is to penetrate biological structures. Therefore, the expected, but unobtained, result would be an increase in reflectance at the lowest wavelength range. UV light is strongly absorbed by melanin as well as blood, so reflectance in this region is significantly affected by their local concentration. An increased concentration of sebum should reduce the reflectance since there is increased absorption in the lower part of the 335–380 nm spectral band.

On the other hand, at the wavelength ranges 400–520 nm and 480–600 nm, a negative correlation was obtained between reflectance and amount of sebum. This indicates that increased sebum secretion causes less radiation to be reflected and thus more to be absorbed/transmitted at this spectral range. If the skin more effectively absorbs radiation at the wavelength range 400–600 nm, the remaining parts of visible radiation will be more "exposed". As a result, it can cause the optical impression of reddened skin. It can therefore be concluded that an increased amount of skin sebum may subjectively cause the skin to seem more red.

For the other spectral ranges, i.e., 590–720 nm, 700–1100 nm, 1000–1700 nm, and 1700–2500 nm, a positive correlation between the amount of sebum and skin reflectance was recorded. This indicates that sebum secretion causes an increase in the radiation energy reflected (diffused) from the surface of the skin. The highest correlation was recorded in the near-infrared range, i.e., 700–1100 nm, and sebum secretion may therefore have a protective effect against infrared radiation. This radiation does not carry as much energy as ultraviolet radiation (radiation wavelength is inversely proportional to energy), but it has a strong negative impact on cells and tissues mainly due to the fact that the depth of skin penetration

increases along with wavelength. Therefore, ultraviolet radiation, despite the fact that it carries a lot of energy, penetrates the skin much shallower than infrared radiation.

A significant difference in the dispersion of results for individual spectral ranges may result from differences in the scattering of radiation through skin unevenness. The longer the wavelength, the less the radiation will be scattered by skin irregularities. The second cause may be the uneven distribution of skin chromophores. The main skin chromophores (hemoglobin and melanin) have absorption maxima in a spectral range between 335 and 720 nm, and their non-homogeneous arrangement may therefore result in the greater dispersion of the results. Thirdly, the penetration of radiation into the skin depends on the wavelength. Higher wavelength radiation penetrates the skin more effectively, which may also contribute to the smaller spread in the reflectance range.

The protection of the skin against infrared radiation from the presence of skin sebum is insignificant. Nevertheless, even this modest protection can have significant physiological and clinical importance in long-term exposure.

It should also be emphasized that many cosmetic and dermatological procedures are associated with the inhibition of sebocyte activity or sebum removal from the skin surface. Skin containing sebum (glossy, oily) is considered unattractive, hence many procedures and cosmetics are designed to reduce sebum secretion or remove it mechanically. However, in light of the obtained results, it seems that sebum, apart from its strictly lubricating effect, probably also has a slight UV-protective effect. Additional research is needed to confirm this hypothesis.

In subsequent research, the authors plan to increase the number of volunteers and also determine the impact of the anatomical location of the measurement and volunteer age, sex, and phototype on the obtained results. An attempt will also be made to assess specular reflectance and diffuse reflectance in order to verify the hypothesis of how surface development (roughness) affects the obtained results.

In conclusion, correlations between sebum content and reflectance were initially determined. Nevertheless, the coefficient is not high and further research is needed to verify the obtained results with a larger number of volunteers of different ages, sex, and phototypes.

## 5. Conclusions

By analyzing the directional reflectance of the skin and the amount of skin sebum, the following conclusions can be drawn:

1. The amount of sebum does not affect the directional reflectance of the skin at the wavelength 335–380 nm;
2. With an increase in the amount of sebum, the directional reflectance of the skin decreases at the wavelengths 400–540 nm and 480–600 nm;
3. With an increase in the amount of sebum, the directional reflectance of the skin increases at the wavelengths 590–720 nm, 700–1100 nm, 1000–1700 nm, and 1700–2500 nm;
4. The closest relationship between amount of sebum and directional reflectance of the skin was observed at the wavelength 700–1100 nm;
5. Sebum may have a weak sunscreen effect in the near-infrared spectral range;
6. The increased absorption of radiation by sebum in the spectral range 400–600 nm may cause skin reddening.

The obtained results are an interesting starting point for continuing research on the effect of sebum content on skin reflectance. Reflecting/scattering radiation from the skin surface depending on sebum content may be clinically significant not only in the context of exposure to solar radiation (which in turn may contribute to photoaging and even skin cancer) but also in the context of numerous therapeutic methods based on artificial sources of radiation. In this area, it is desirable for radiation to penetrate the skin as effectively as possible. The obtained preliminary results confirm that the used method is an interesting alternative to spectroscopic methods.

**Author Contributions:** Conceptualization, A.B., K.K., M.K. and S.W.; methodology, A.B., M.H.-P., B.K.-Ś. and K.K.; software, P.P., M.H.-P. and S.W.; validation, A.B. and M.H.-P.; formal analysis, A.B., M.K. and B.K.-Ś.; investigation, A.B., M.K. and K.K.; resources, S.W.; data curation, A.B., M.H.-P. and K.K.; writing—original draft preparation, A.B., K.K., A.L. and M.H.-P.; writing—review and editing, A.B., A.L. and S.W.; visualization, A.B. and M.H.-P.; supervision, S.W.; project administration, A.B. and S.W.; funding acquisition, S.W. All authors have read and agreed to the published version of the manuscript.

**Funding:** This research was funded by the Medical University of Silesia, grant numbers PCN-1-128/N/2/O; PCN-1-057/K/2/O.

**Institutional Review Board Statement:** This study was conducted in accordance with the Declaration of Helsinki and approved by the Bioethics Committee of the Medical University of Silesia (PCN/CBN/0022/KB1/27/III/16/17/21).

**Informed Consent Statement:** Informed consent was obtained from all subjects involved in the study.

**Data Availability Statement:** The datasets used and/or analyzed during the current study are available from the corresponding author on reasonable request.

**Conflicts of Interest:** The authors declare no conflict of interest.

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
