# Peer review of "The Influence of Sebum on Directional Reflectance of the Skin"

_applsci, doi:10.3390/app13052838_

Round 1

Reviewer 1 Report

In this article, the author proposed the relationship between the sebum and directional reflectance of the skin. It is meaningful research. As a researcher on EE, we think some important parameters should be supplied.  

The size, reflectance, and thickness of the sebum should be supplied, which are all important to the reflectance of the system.

Author Response

Anna Banyś, Ph.D.                                                                              Sosnowiec, February 13th 2023

Department of Basic Biomedical Science,

Faculty of Pharmaceutical Sciences in Sosnowiec,                                           

Medical University of Silesia in Katowice, Poland

Dear Reviewer,

I am very grateful for the valuable opinions and remarks. I agree with all the comments and I have referred to them as best as possible in a revised version of the manuscript. List of corrections is presented below:

  • In this article, the author proposed the relationship between the sebum and directional reflectance of the skin. It is meaningful research. As a researcher on EE, we think some important parameters should be supplied.  

The size, reflectance, and thickness of the sebum should be supplied, which are all important to the reflectance of the system.

ANSWER: The studies were conducted in vivo, so it is not possible to provide standardized parameters of sebum content on the skin. Nevertheless, the results obtained with the new, proposed method, i.e. with the use of hemispheric directional reflectance, were correlated with the gold standard of sebum content determination method, i.e. the sebumeter. The sebumeter was calibrated before each measurement.

Reviewer 2 Report

The focus of the manuscript is comparison of two measurement approaches of skin sebum concentration. Specifically, by a standard sebumeter and an alternative technique solar reflectometer. My main concern is the use of a non-standard skin spectrometry approach. A spectrometer and integrating sphere would be more suitable for such study, since comparison to the other studies would be easier and also interpretation of the obtained results would be more accurate. The second concern is the group of volunteers. The authors do not report how many of them were males and how many females, what was their skin type, did they all have comparable diet, etc. Also, the age of the volunteers is too narrow. I’d suggest the authors to include also older population and different skin types if they did not include them already. The third concern is that all measurements were averaged together. It’d be good to analyze the measurements by body regions and genders. I’d assume that the measurement points spread could be reduced in case of subgroups, since skin of the volunteers is more similar in the same regions.

Specific comments are below.

Abstract: Some of the information provided in the introductory part are not crucial for understanding the study (e.g., EM radiation, cancer, vitamin D), so I’d suggest keeping only the part crucial for the study. Also, some numbers would be appreciated regarding the sebum increase – directional reflectance increase. A conclusion sentence is missing, while details (e.g., skin reddening, sunscreen effect) should be removed.

Introduction:

·         The motivation for the study is not clear from the introduction and therefore it should be clearly stated. Stating only that the goal of the study was to measure skin reflectance in specific spectral region is not enough. Please provide the reader an explanation why your study is important for the field and what open question it addresses.

·         The review of the filed regarding sebum skin concentration, skin reflectance detection, techniques for measuring sebum layer thickness is completely missing. Please provide the missing information.

·         Certain statements are misleading if not completely wrong (e.g. “the more hydrated the skin, the smoother it is and the better the reflection of the light beam, and thus the higher the reflectance values are” – not true, air pockets and densely packed fibers in dry skin increase skin scattering and thus diffuse reflectance). Please read the introduction carefully and support all the statements with trustworthy references.

Materials and Methods

·         Information about the volunteers is missing how many males and how many females, what was their skin type, was the skin tanned, when the measurement was performed (time of the day, month), what was the humidity and temperature, did they wash skin with soap few hours before the measurement, did the volunteers rest before the measurements, etc. Why such narrow age interval was selected? Skin of older people differs significantly from skin of the young ones.

·         Ls. 59–60: “The obtained results are the average of all measurements, and were the subject of further analysis.” What does this statement mean? That the results are average of the three/two measurements per spot or all measurements for specific region or for all volunteers? In the latter two cases the analysis is wrong. I also wonder why two/three measurements were performed? Was the reputability that good?

·         When introducing an instruments or a software, provide also info about the manufacturer and the county in addition to the instrument model or software version.

·         Ls. 67–73: Commonly integrating sphere coupled to a spectrometer is used for skin spectrometry and the result is either a diffuse or total reflectance. Why a device for inspecting solar reflectance/absorbance and mirror inspection was used instead? Since the approach is not a standard one, the authors should explain in the detail, how the instrument works (e.g., incidence angle, light source, integration time, the accuracy of the measurements, calibration) and what are the measurement results (following from the incident and detected light intensity). How was the pressure of the instrument on the skin controlled?

Results

·         Figure 1: How amount of sebum was measured? I guess by sebumeter considering the units. Why were the presented bins selected? A report about how different body sides differed in sebum concentration and difference between male and female subjects would be very informative.

·         Figure 2: The results are expected considering the ordinary skin and sebum reflectance spectra. What should be discussed is why specific spectral bands have a large spread of the values (e.g. 335-380 nm), while other almost negligible (e.g. 1700-2500 nm). This can be done based on the absorbance and scattering spectrum of human skin and sebum in these spectral regions. A comparison between different body sites and genders would be appreciated. Table 1 does not provide any additional information and should be removed.

·         Figs. 3–9 are all very similar. I’d suggest showing only the most relevant ones and explain the rest in the text. Visually, the spread of the measurement points for all spectral bands is significant and R’s are relatively small showing that such simple comparison of the measurements obtained by two completely different instruments is inadequate. The authors should explore the origin of this discrepancy (e.g., measurement sites, gender, skin type, skin cleaning, diet, resting state, number of volunteers).

Discussion

·         Ls. 128–145 – The purpose of this study is not to explore the (pato)physiological and pathological effect of sebum on the skin, but a mere comparison of the two different approaches to measure sebum concentration. The unnecessary information should be removed from the discussion.

·         Ls. 146–150 – This should be part of the Introduction including a brief explanation of the technique and relevant references.

·         Many of the explanations for the observed trends (or missing trends) are only hypotheses without a proper scientific support. I’ll explain some of them, but the authors should put effort into providing adequate explanation for all their hypotheses, preferably referring to skin physiology, morphology and optical properties of skin constituents and sebum.

·         Ls. 160–161: “Therefore, the expected, but unobtained result would be an increase in reflectance at the lowest wavelength range.” The UV light is strongly absorbed in melanin and also blood, so the reflectance in this region is significantly affected by their local concentration. The increased concentration of sebum should reduce the reflectance, since there is increased absorption in the lower part of the 335-380 nm spectral band.

·         Ls. 162–169: “On the other hand, at the wavelength ranges 400 - 520 nm and 480 - 600 nm, a negative correlation was obtained between reflectance and the amount of sebum”– Sebum is not expected to have a significant absorption in this range (lipids), except if there are different carotenoids due to individual diet present.
“As a result, it can cause the optical impression of reddened skin.” – Was this actually observed? I’d assume that higher concentrations of sebum were observed in regions where skin is more perfused and correspondingly reddish.

·         Ls. 170–180: “For the other spectral ranges, i.e. 590-720 nm, 700-1100 nm, 1000-1700 nm and 1700–2500 nm, a positive correlation between the amount of sebum and skin reflectance was

·recorded.” – This is a surprising finding. According to the literature the pure sebum should have larger absorption coefficient in the SWIR region (900-3000 nm) compared to water (the main skin component in this spectral region), thus I’d expect a negative trend in the latter three spectral bands. There must be another reason for the observed positive trend.

·         Comparison of this study results to other studies reporting skin spectroscopy with or without sebum is missing. How relevant this study findings are?

·         Future plans and study shortcomings are missing in the discussion.

·         Ls. 191–192: “In conclusion, a correlation between the amount of sebum and reflectance of the

·skin was found.” This statement should be mitigated, since R’s were not large.

Conclusions

The conclusions are not in a standard form. And all this was already presented in the results and discussion. The conclusions should be a wrap-up of the study including the outlook.

Author Response

Anna Banyś, Ph.D.                                                                             Sosnowiec, February 13th 2023

Department of Basic Biomedical Science,

Faculty of Pharmaceutical Sciences in Sosnowiec,                                           

Medical University of Silesia in Katowice, Poland

Dear Reviewer,

I am very grateful for the valuable opinions and remarks. I agree with all the comments and I have referred to them as best as possible in a revised version of the manuscript. List of corrections is presented below:

The focus of the manuscript is comparison of two measurement approaches of skin sebum concentration. Specifically, by a standard sebumeter and an alternative technique solar reflectometer. My main concern is the use of a non-standard skin spectrometry approach. A spectrometer and integrating sphere would be more suitable for such study, since comparison to the other studies would be easier and also interpretation of the obtained results would be more accurate. The second concern is the group of volunteers. The authors do not report how many of them were males and how many females, what was their skin type, did they all have comparable diet, etc. Also, the age of the volunteers is too narrow. I’d suggest the authors to include also older population and different skin types if they did not include them already. The third concern is that all measurements were averaged together. It’d be good to analyze the measurements by body regions and genders. I’d assume that the measurement points spread could be reduced in case of subgroups, since skin of the volunteers is more similar in the same regions.

Specific comments are below.

Abstract: Some of the information provided in the introductory part are not crucial for understanding the study (e.g., EM radiation, cancer, vitamin D), so I’d suggest keeping only the part crucial for the study. Also, some numbers would be appreciated regarding the sebum increase – directional reflectance increase. A conclusion sentence is missing, while details (e.g., skin reddening, sunscreen effect) should be removed.

ANSWER: Abstract has been changed as suggested.

Introduction:

  • The motivation for the study is not clear from the introduction and therefore it should be clearly stated. Stating only that the goal of the study was to measure skin reflectance in specific spectral region is not enough. Please provide the reader an explanation why your study is important for the field and what open question it addresses.

ANSWER: The explanation has been added: “The correlation between sebum content and reflectance value in the studied spectral range may have significant practical implications. First of all, increasing of the reflectance of skin with sebum vs skin without sebum may be important in the skin imaging and in the case of use all kind of radiation sources affecting the skin. This applies especially to dermatology and aesthetic dermatology. The increase in the reflection coefficient for skin covered with sebum indicates that, for example, laser treatments in a given spectral range should be preceded by skin degreasing. In turn, a decrease in the reflection coefficient could indicate that the radiation penetrates the skin more effectively, and thus its dose should be optimized in relation to the sebum content. At the same time, sebum, by scattering radiation, can change the color rendering index and thus affect all skin image acquisition techniques in the examined spectral range. Therefore, understanding the correlation between the sebum content and the radiation scattering factor is also important from a clinical point of view.”

The review of the filed regarding sebum skin concentration, skin reflectance detection, techniques for measuring sebum layer thickness is completely missing. Please provide the missing information.

ANSWER: The missing information has been provided:

“With almost normal (nearly perpendicular) radiation on the skin, only a small amount is reflected due to the change in refractive index, which is 1.0 for air and about 1.55 for stratum corneum. Therefore, in the case of perpendicularly incident radiation in the spectral range of 250 - 3000 nm, about 4-7% of the radiation is reflected/scattered (regardless of the melanin content in the skin). However, at an angle of incidence other than 90 degrees, as is the case in practice, a much higher percentage of the radiation is reflected/scattered from the skin. Because the surface of the stratum corneum is not smooth and flat, the skin reflectance is not specular. The reflectance, that occurs on the surface of the skin, can be clinically significant. Significantly, sebum may be an important factor influencing the air-tissue reflectance/dispersion coefficient. Sebum can act not only as a combining factor, but also as a “smoothing” factor for the stratum corneum [11].

Sebum is a substance that occurs naturally on the surface of the epidermis and is produced by the sebaceous glands. The main function of sebum is to moisturize, lubricate and protect the skin and hair, maintaining an effective hydrophobic barrier, preventing water loss and the invasion of microorganisms [12]. The composition and rate of sebum production varies from person to person and depends on where on the body sebum is produced. Sebum is a complex mixture of glycerides, free fatty acids, wax esters and squalene. The amount and dynamics of sebum secretion change in response to numerous environmental factors, including seasons [12, 13]. Sebum production increases in summer. In addition, androgens in men stimulate especially the growth of sebum-producing sebaceous glands, and these are located primarily on the face, chest and upper back. Systemic medications that increase sebum secretion include: testosterone, progesterone and phenothiazine. Among the drugs that reduce the secretion of sebum, retinoids should be mentioned first [14].

The gold standard in the assessment of sebum content is the use of the method of absorption by tapes that change their optical properties after contact with sebum. The light transmittance of the absorbing tape is measured after it is in contact with epidermis with sebum on the surface. The absorbent tape becomes transparent when in contact with sebum on the surface of the skin. To assess the amount of sebum, the applicator is inserted into the main unit where the transparency of the film is measured by a light source. The amount of light that passes through the tape is measured by a photocell. The light penetration reflects the sebum content on the surface of the skin. The measuring surface in the case of a sebumeter is about 64 mm2, the measurement accuracy is 5%.”

Certain statements are misleading if not completely wrong (e.g. “the more hydrated the skin, the smoother it is and the better the reflection of the light beam, and thus the higher the reflectance values are” – not true, air pockets and densely packed fibers in dry skin increase skin scattering and thus diffuse reflectance). Please read the introduction carefully and support all the statements with trustworthy references.

ANSWER: Supplemented with: “The reflectance/diffuse of the skin depends on the water content but also on the smoothness of the surface. Moisturizing the skin, on the one hand, causes a decrease in the refractive index (more radiation penetrates the skin), but on the other hand, the increase in hydration causes the smoothness of the skin. Therefore, it is to be expected that skin hydration should increase the specular reflectance but may decrease the diffuse reflectance.”

Materials and Methods

Information about the volunteers is missing how many males and how many females, what was their skin type, was the skin tanned, when the measurement was performed (time of the day, month), what was the humidity and temperature, did they wash skin with soap few hours before the measurement, did the volunteers rest before the measurements, etc. Why such narrow age interval was selected? Skin of older people differs significantly from skin of the young ones.

ANSWER: We fully agree with the Reviewer's comment that the size and homogeneity of the population could have affected the results. Nevertheless, the idea behind the study - as a preliminary study - was to indicate that the use of hemispheric directional reflectance can be a very promising method in assessing the correlation between sebum content and skin reflectance. In subsequent studies, we plan to build a database of volunteers based on gender, phototype, age and seborrheic skin diseases.

Added: “All volunteers are women characterised by II and III skin phototype. The volunteers did not sunbathe for at least 2 months before the study and did not use any cosmetics on the day of the study. The research was conducted in June 2022. The temperature and humidity of the rooms were controlled and amounted to: temperature 21oC, humidity 40%.”

Ls. 59–60: “The obtained results are the average of all measurements, and were the subject of further analysis.” What does this statement mean? That the results are average of the three/two measurements per spot or all measurements for specific region or for all volunteers? In the latter two cases the analysis is wrong. I also wonder why two/three measurements were performed? Was the reputability that good?

ANSWER: To increase the reliability of the measurements, with a small number of volunteers, the methods were validated based on the repeatability of the measurement, both for the sebumeter and the reflectometer. The repeatability of the results for the reflectometer was very high - a difference of less than 1%. For the sebumeter, this value was below 5%. The relatively low repeatability for the sebumeter is due to the measurement concept. The absorbent tape cannot be applied twice to the same place because the first measurement should absorb all the sebum. Therefore, attempts were made to apply the tape as close as possible (but not tangentially) to the place of the previous measurement.

When introducing an instruments or a software, provide also info about the manufacturer and the county in addition to the instrument model or software version.

ANSWER: Provided.

Ls. 67–73: Commonly integrating sphere coupled to a spectrometer is used for skin spectrometry and the result is either a diffuse or total reflectance. Why a device for inspecting solar reflectance/absorbance and mirror inspection was used instead? Since the approach is not a standard one, the authors should explain in the detail, how the instrument works (e.g., incidence angle, light source, integration time, the accuracy of the measurements, calibration) and what are the measurement results (following from the incident and detected light intensity). How was the pressure of the instrument on the skin controlled?

ANSWER: The additional information have been provided: “In this study the total reflectance was analyzed. A reflectometer that works in the spectral range of 335 - 2500 nm, with an incidence angle of 60 degrees was used, be-cause it faithfully reflects skin exposure to natural solar radiation. In addition, the vast majority of radiation emitters (lasers, IPL, diode lamps and others) work in this spec-tral range, because it covers the full range of the so-called absorption maxima for the main skin chromophores, i.e. water, hemoglobin and melanin.”

Results

  • Figure 1: How amount of sebum was measured? I guess by sebumeter considering the units. Why were the presented bins selected? A report about how different body sides differed in sebum concentration and difference between male and female subjects would be very informative.

      ANSWER: Sebumetric measurements do not have units. It is a relative unit. We focus on the measurement of sebum on the face because the activity of the sebaceous glands is highest in the face. In subsequent studies, we plan to extend the results to include more volunteers, of different sexes and phototypes.

      Figure 2: The results are expected considering the ordinary skin and sebum reflectance spectra. What should be discussed is why specific spectral bands have a large spread of the values (e.g. 335-380 nm), while other almost negligible (e.g. 1700-2500 nm). This can be done based on the absorbance and scattering spectrum of human skin and sebum in these spectral regions. A comparison between different body sites and genders would be appreciated. Table 1 does not provide any additional information and should be removed.

      ANSWER: Table 1 has been deleted.

      Added: “A significant difference in the dispersion of results for individual spectral ranges may resulted from a differences in the scattering of radiation on skin unevenness. The longer the wavelength, the less the radiation will be scattered on skin irregularities. The second cause may be the uneven distribution of skin chromophores. The main skin chromophores (hemoglobin and melanin) have absorption maxima in the spectral range between 335 and 720 nm, therefore their non-homogeneous arrangement may result in a greater dispersion of results. Thirdly, the penetration of radiation into the skin depends on the wavelength. Higher wavelength radiation penetrates the skin more effectively, which may also contribute to a smaller spread in the reflectance range.”

      Figs. 3–9 are all very similar. I’d suggest showing only the most relevant ones and explain the rest in the text. Visually, the spread of the measurement points for all spectral bands is significant and R’s are relatively small showing that such simple comparison of the measurements obtained by two completely different instruments is inadequate. The authors should explore the origin of this discrepancy (e.g., measurement sites, gender, skin type, skin cleaning, diet, resting state, number of volunteers).

      ANSWER: The idea of the work was preliminary study. The authors are aware of the limits of the method. Nevertheless, in subsequent works, they will try to systematically expand the options (number of volunteers, gender, age, etc.), which will contribute to an even better understanding of the observed relations.

Discussion

  • Ls. 128–145 – The purpose of this study is not to explore the (pato)physiological and pathological effect of sebum on the skin, but a mere comparison of the two different approaches to measure sebum concentration. The unnecessary information should be removed from the discussion.
  • Ls. 146–150 – This should be part of the Introduction including a brief explanation of the technique and relevant references.

ANSWER: It has been delated.

  • Many of the explanations for the observed trends (or missing trends) are only hypotheses without a proper scientific support. I’ll explain some of them, but the authors should put effort into providing adequate explanation for all their hypotheses, preferably referring to skin physiology, morphology and optical properties of skin constituents and sebum.
  • Ls. 160–161: “Therefore, the expected, but unobtained result would be an increase in reflectance at the lowest wavelength range.” The UV light is strongly absorbed in melanin and also blood, so the reflectance in this region is significantly affected by their local concentration. The increased concentration of sebum should reduce the reflectance, since there is increased absorption in the lower part of the 335-380 nm spectral band.

      ANSWER: Thank you very much for this statement. We supplemented the text of the manuscript with:

“The UV light is strongly absorbed in melanin and also blood, so the reflectance in this region is significantly affected by their local concentration. The increased concentration of sebum should reduce the reflectance, since there is increased absorption in the lower part of the 335-380 nm spectral band.”

  • Ls. 162–169: “On the other hand, at the wavelength ranges 400 - 520 nm and 480 - 600 nm, a negative correlation was obtained between reflectance and the amount of sebum”– Sebum is not expected to have a significant absorption in this range (lipids), except if there are different carotenoids due to individual diet present.
    “As a result, it can cause the optical impression of reddened skin.” – Was this actually observed? I’d assume that higher concentrations of sebum were observed in regions where skin is more perfused and correspondingly reddish.

      ANSWER: Measurements of sebum content actually included anatomical areas exposed to telangiectasias. However, none of the volunteers had visible dilated blood vessels or rosacea. The criterion for exclusion from the study was any skin disorder or dermatoses.

      The idea behind such a conclusion was as follows: the greater the difference in the absorption/scattering of radiation in the visible light range by sebum, the greater the effect on the observed skin color. Thus, if sebum causes an increased absorption of radiation in a given spectral range, the greater the percentage of the other spectral range, which may affect the subjective perception of skin color.

       Ls. 170–180: “For the other spectral ranges, i.e. 590-720 nm, 700-1100 nm, 1000-1700 nm and 1700–2500 nm, a positive correlation between the amount of sebum and skin reflectance was·recorded.” – This is a surprising finding. According to the literature the pure sebum should have larger absorption coefficient in the SWIR region (900-3000 nm) compared to water (the main skin component in this spectral region), thus I’d expect a negative trend in the latter three spectral bands. There must be another reason for the observed positive trend.

      ANSWER: A probable explanation for this trend is the "smoothing" of the skin surface. Sebum absorption studies were conducted in vitro, so this effect has not been observed yet. The next step of the research will be to determine not only total reflectance, but also diffuse reflectance and specular reflectance of the skin in vivo, which will probably help explain this effect.

  • Comparison of this study results to other studies reporting skin spectroscopy with or without sebum is missing. How relevant this study findings are?

ANSWER: The results were not compared with other studies, because no studies were found that included the assessment of absorption/reflectance of sebum of the skin in vivo. Nevertheless, attempts have been made to explain the obtained results based on the physiology of the skin and the optical properties of the skin.

Future plans and study shortcomings are missing in the discussion.

      ANSWER: Added: “In subsequent works, the authors plan to increase the number of volunteers, but also determine the impact of the anatomical location of the measurement, volunteers' age, volunteers' sex, and volunteers' phototype on the obtained results. An attempt will also be made to assess specular reflectance and diffuse reflectance in order to verify the hypothesis of how surface development (roughness) affects the obtained results.”

  • Ls. 191–192: “In conclusion, a correlation between the amount of sebum and reflectance of the skin was found.” This statement should be mitigated, since R’s were not large.

ANSWER: Changed the sentences to:

“In conclusion, correlations between sebum content and reflectance were initially determined. Nevertheless, this coefficient is not high and further research is needed to verify the obtained results with the larger number of volunteers (different age, sex, phototype).”

Conclusions

The conclusions are not in a standard form. And all this was already presented in the results and discussion. The conclusions should be a wrap-up of the study including the outlook.

ANSWER: The wrap-up paragraph has been added: “The obtained results are an interesting starting point for continuing research on the effect of sebum content on skin reflectance. Reflecting/scattering radiation from the skin surface, depending on the sebum content, may not only be clinically significant in the context of exposure to solar radiation (which in turn may contribute to photoaging and even skin cancer) but also in the context of numerous therapeutic methods based on artificial sources of radiation. In this area, we want the radiation to penetrate the skin as effectively as possible. The obtained preliminary results confirm that the method used may be an interesting alternative to spectroscopic methods.”

Reviewer 3 Report

I feel the number of subjects is not enough to support the conclusion.  Also, the authors should explain why only people aged 23-25 were recruited for this study.  The findings could be more meaningful and robust with a broader age range and more subjects.  

- It doesn't look like the histogram of the amount of skin sebum (Figure 1) follows the normal distribution.  The authors should not use the normal distribution to describe the data.

Author Response

Anna Banyś, Ph.D.                                                                             Sosnowiec, February 13th 2023

Department of Basic Biomedical Science,

Faculty of Pharmaceutical Sciences in Sosnowiec,                                           

Medical University of Silesia in Katowice, Poland

Dear Reviewer,

I am very grateful for the valuable opinions and remarks. I agree with all the comments and I have referred to them as best as possible in a revised version of the manuscript. List of corrections is presented below:

  • I feel the number of subjects is not enough to support the conclusion. Also, the authors should explain why only people aged 23-25 were recruited for this study.  The findings could be more meaningful and robust with a broader age range and more subjects.

It doesn't look like the histogram of the amount of skin sebum (Figure 1) follows the normal distribution.  The authors should not use the normal distribution to describe the data.

ANSWER: In subsequent studies, the authors plan to study more volunteers. Additional groups of volunteers of different ages and phototypes will be created to determine the influence of skin chromophores on the obtained results.

The normality of the distribution was confirmed by the Shapiro-Wilk test. Friedman's ANOVA test and Dunn's post-hoc test were used to assess the differences in directional reflectance of the skin at different wavelength ranges.

Round 2

Reviewer 1 Report

The reflectance of sebum may be searched by the material.

The thickness of the sebum can be measured by optical method.

Reviewer 3 Report

My concerns have been resolved by the authors, and I don't have any further comments.